# Prenatal and Postnatal Methyl-Modulator Intervention Corrects the Stress-Induced Glucocorticoid Response in Low-Birthweight Rats

**DOI:** 10.3390/ijms22189767

**Published:** 2021-09-09

**Authors:** Takahiro Nemoto, Yoshihiko Kakinuma

**Affiliations:** Department of Bioregulatory Science (Physiology), Nippon Medical School, Tokyo 113-8602, Japan; k12417853@nms.ac.jp

**Keywords:** stress response, glucocorticoid, long noncoding RNA, pituitary, nutrition

## Abstract

Low body weight at birth has been shown to be a risk factor for future metabolic disorders, as well as stress response abnormalities and depression. We showed that low-birthweight rats had prolonged high blood corticosterone levels after stress exposure, and that an increase in Gas5 lncRNA, a decoy receptor for glucocorticoid receptors (GRs), reduces glucocorticoid responsiveness. Thus, we concluded that dampened pituitary glucocorticoid responsiveness disturbed the glucocorticoid feedback loop in low-birthweight rats. However, it remains unclear whether such glucocorticoid responsiveness is suppressed solely in the pituitary or systemically. The expression of Gas5 lncRNA increased only in the pituitary, and the intact induction of expression of the GR co-chaperone factor Fkbp5 against dexamethasone was seen in the liver, muscle, and adipose tissue. Intervention with a methyl-modulator diet (folate, VB_12_, choline, betaine, and zinc) immediately before or one week after delivery reversed the expression level of Gas5 lncRNA in the pituitary of the offspring. Consequently, it partially normalized the blood corticosterone levels after restraint stress exposure. In conclusion, the mode of glucocorticoid response in low-birthweight rats is impaired solely in the pituitary, and intervention with methyl-modulators ameliorates the impairment, but with a narrow therapeutic time window.

## 1. Introduction

Factors that determine the stress responsiveness of an infant are related not only to the growing environment in infancy (i.e., maternal care), but also to the nutritional or hygienic environment in the embryonic period. In fact, epidemiological findings of the Dutch famine showed that low-birthweight (LBW) infants from pregnant women who faced famine in the early stages of pregnancy had a higher risk of stress response abnormalities, cognitive decline, and depression, as well as various metabolic and cardiovascular diseases [1,2,3,4]. The response to stress is characterized by activation of the hypothalamus–pituitary–adrenal (HPA) axis and the sympathetic adrenomedullary system [5]. Cortisol suppresses unnecessary or harmful functions, especially in fight-or-flight situations [6]. It alters the response of the immune system [7] and suppresses digestive [8], reproductive [9], and growth [10] processes. This complex natural alert system plays a critical role in communicating with various areas of the brain that control mood [11], motivation [12], and fear [13]. We created a rat model to elucidate the mechanism by which embryonic nutrition affects the stress response [14]. LBW rats with embryonic malnutrition had higher levels of Gas5 lncRNA expression in the pituitary gland than control rats. Gas5 lncRNA is a long noncoding RNA that has a sequence similar to the glucocorticoid response element sequence, and it functions as a decoy receptor for the glucocorticoid receptor (GR) [15]. In the pituitary gland of LBW rats, we concluded that GR could not bind to DNA due to the increased expression of Gas5 lncRNA, and glucocorticoid responsiveness was impaired. Moreover, the impaired glucocorticoid responsiveness reduced the induction of expression of miR-449a, whose expression is induced by glucocorticoids [16]. miR-449a is a microRNA that suppresses the expression of the corticotropin-releasing factor receptor (CRF-R1) and is a factor that plays a part in the negative feedback mechanism of glucocorticoids in the pituitary gland [17].

In Japan, the mean birth weight has decreased over time, and the birth rate of LBW infants has been increasing [18]. Therefore, the development of preemptive medicine is an urgent issue to counter the increasing number of LBW infants. The goal of preemptive medicine is to identify children with a high risk of diseases early in life and adopt a preventive intervention to retard disease onset. We attempted to feed a diet enriched with folic acid and vitamin B_12_, which also functions as an epigenetic modulator, to correct dysregulated gene expression in model rats [19]. However, to assess the effects of nutritional interventions, it was necessary to clarify whether abnormal glucocorticoid responsiveness in LBW rats was confined to the pituitary gland or systemically impaired. Therefore, we first investigated the expression of Gas5 lncRNA, which causes abnormal glucocorticoid responsiveness in GR-expressing peripheral organs, e.g., the liver, skeletal muscle, and adipose tissue, in LBW rats and control normal birthweight (NBW) rats, and investigated the glucocorticoid response after administration of dexamethasone through examining the expression level of Fkbp5 mRNA. Fkbp5 is a co-chaperone factor of GR, and it has been reported that its expression level increases in response to glucocorticoids [20]. Therefore, the expression levels in various organs of dexamethasone-treated rats were compared with those of saline-administered rats. Subsequently, the glucocorticoid responsiveness with methyl modulator supplementation was evaluated. Since the critical window for administration of methyl modulators was unknown, the intervention was performed before and after birth and after weaning.

## 2. Results

### 2.1. Comparison of Gas5 lncRNA Expression Levels

The expression level of Gas5 lncRNA in the pituitary of LBW rats was significantly increased compared with that of NBW rats (*p* < 0.0001, *n* = 7, Figure 1A). However, there were no significant differences in the expression levels in the liver (Figure 1B), skeletal muscle (Figure 1C), and adipose tissue (Figure 1D) between LBW and NBW rats.

### 2.2. Comparison of Fkbp5 Expression Levels in Response to Dexamethasone

In NBW rats, dexamethasone administration significantly increased Fkbp5 mRNA expression in the pituitary (*p* < 0.01, *n* = 6, Figure 2A). However, there was no significant difference in the dexamethasone-induced expression level of Fkbp5 mRNA in the pituitary of LBW rats (Figure 2A). In the liver, dexamethasone-induced Fkbp5 mRNA expression was significantly increased in both NBW (*p* < 0.001, *n* = 6) and LBW (*p* < 0.01, *n* = 6) rats (Figure 2B). Similarly, dexamethasone-induced Fkbp5 mRNA was significantly increased in skeletal muscle (Figure 2C) and adipose tissue (Figure 2D).

### 2.3. Normalization of Gas5 lncRNA Expression in the Pituitary by Methyl Modulator Nutritional Intervention

As shown in Figure 3, rats were fed with a methyl modulator diet at various times: for three days of lactation immediately after delivery; for one week of lactation immediately after delivery; in the third trimester of gestation; and for 1 week post-weaning. The expression level of Gas5 lncRNA in the pituitary of the offspring from the methyl modulator feeding dams for 3 days immediately after delivery (LBW-methyl-3d) was not significantly different from that of non-intervention LBW rats. When the methyl modulator was administered to dams for 1 week immediately after delivery (LBW-methyl-1w), the Gas5 lncRNA levels in the pituitary of the offspring was significantly reduced compared to that in the non-intervention (*p* < 0.0001, *n* = 6), and its expression level was comparable with that in NBW rats (Figure 4). In addition, feeding methyl modulators to dams in the third trimester (LBW-prenatal-methyl) significantly reduced Gas5 lncRNA expression levels in the pituitary of the offspring compared to non-intervention (*p* < 0.0001, *n* = 6). However, when the methyl modulator was fed to the offspring after weaning (LBW-postweaning-methyl), the expression level of Gas5 lncRNA in the pituitary remained significantly higher than that of NBW (*p* < 0.0001, *n* = 6), and no significant difference was seen in the expression level with non-intervention (Figure 4).

### 2.4. Effect of Methyl Modulator Nutritional Intervention on Stress-Induced Corticosterone Levels

Blood corticosterone levels in NBW rats subjected to restraint stress returned to a basal level and were not significantly different after 120 min restraint stress (Figure 5), whereas the levels at 120 min stress in LBW (non-intervention) rats were significantly higher than basal levels (*p* < 0.0001, *n* = 6) and significantly higher than of NBW rats (*p* < 0.01, n = 6). LBW-methyl-3d also significantly increased the levels, but with no significant difference from non-intervention, but blood corticosterone levels after restraint stress were significantly lower in offspring LBW-methyl-1w immediately after birth than in non-intervention (*p* < 0.05, *n* = 6). Similarly, the nutritional intervention of LBW-prenatal-methyl significantly reduced offspring’s blood corticosterone levels after restraint stress exposure compared to non-intervention (*p* < 0.0001, *n* = 6). However, the intervention of methyl modulators in LBW-postweaning-methyl failed to lower blood corticosterone levels after stress exposure.

## 3. Discussion

When an individual is exposed to stress, the hypothalamus releases a corticotropin-releasing factor, and the corticotroph of the anterior pituitary that receives it secretes adrenocorticotropin, resulting in increased cortisol/corticosterone secretion from the adrenal cortex. Glucocorticoid suppresses the hormone secretion from the hypothalamus and pituitary gland by a negative feedback regulatory mechanism to maintain a constant blood level. Glucocorticoid receptors regulate gene expression involved in the stress response, development, metabolism, and immune activity [21,22,23,24]. Therefore, dysregulation of glucocorticoid levels is involved in the pathophysiology of various diseases, including psycho-neuronal diseases. For example, alterations in glucocorticoid responsiveness have been associated with major depressive disorder and some anxiety disorders [25]. Elucidation of a control mechanism of glucocorticoid responsiveness is an important issue to help us understand the pathophysiology of diseases, but there are many unclear points. We have previously shown that LBW rat offspring have higher blood corticosterone levels after restraint stress exposure [14]. We also reported that upregulation of Gas5 lncRNA in the anterior pituitary of LBW rats is involved in prolonged elevation of blood corticosterone levels after restraint stress. In other words, we concluded that LBW rats had decreased glucocorticoid responsiveness due to overexpression of Gas5 lncRNA in the pituitary, resulting in sustained high blood corticosterone levels due to an impaired negative feedback regulatory mechanism. In the present study, upregulation of Gas5 lncRNA was found to occur only in the pituitary; it was not observed in the liver, skeletal muscle, or adipose tissues that express GR. In our previous pilot study, we assumed that DNA methylation of Gas5 lncRNA was responsible for the upregulation, and examined DNA methylation in the coding region, but there was no difference from control rats. Elevated expression of Gas5 lncRNA has been reported to occur in triple-negative breast cancer cells [26], polycystic ovary syndrome [27], osteoporosis [28], and liver cancer [29]. However, details of the regulatory expression mechanism of Gas5 lncRNA still remain unknown, and it is not clear why Gas5 lncRNA is expressed in an organ-specific manner. The details of the cell distribution in the pituitary gland and the distribution in the brain of Gas5 lncRNA are not yet clear, but LNCExp database searches show that Gas5 lncRNA is ubiquitously expressed in various organs and cell types, including the brain. Wu et al. reported that silencing of lncRNA GAS5 could activate PI3K/AKT pathway to protect hippocampal neurons against damage in mice with depression-like behaviors by regulating the miR-26a/EGR1 axis [30]. Their results indicate that Gas5 lncRNA not only acts as a decoy receptor for glucocorticoid receptor but also may inhibit intracellular signal transduction via miRNA. We are planning an experiment to establish an in situ hybridization detection system for Gas5 lncRNA, analyze the expression level of Gas5 lncRNA in the brain in low birth weight rats, and clarify its involvement in neuropsychiatric disorder-like behavioral changes seen in low-birthweight rats.

Fkbp5 plays an inhibitory role in GR signaling. Among multiple interacting molecules with Fkbp5, the most important for stress regulation is heat shock protein 90 and other co-chaperones in the steroid receptor complex [31]. By interacting with this complex, Fkbp5 can regulate the sensitivity of GR. That is, Fkbp5 interrupts the interaction of the GR complex with the transport protein dynein, attenuates nuclear translocation, and downregulates GR-dependent transcriptional activity [32]. However, when the ligand binds, Fkbp5 is replaced by Fkbp4, a co-chaperone that recruits dynein to the GR complex and promotes nuclear translocation and transcriptional regulation [33]. GR activation rapidly inhibits genes encoding HPA axis mediators such as CRF and ACTH. On the other hand, activation of GR in multiple tissues rapidly induces transcription and translation of Fkbp5 [34], and then, Fkbp5 inhibits GR activity [32]. Thus, GR-mediated Fkbp5 induction generates an intracellular ultra-short negative feedback loop that regulates HPA activity. However, the induction level of Fkpb5 mRNA may vary across individuals and has been proposed as a marker for GR sensitivity [35,36]. After stimulation with dexamethasone or stress exposure, Fkbp5 mRNA expression is dramatically increased in many brain regions, with the largest changes observed in the amygdala and paraventricular nucleus, but less in the hippocampus [37]. In other words, GR sensitivity can be measured by comparing the expression of Fkbp5 mRNA. In the present study, the expression of Gas5 lncRNA, which inhibits GR binding to GRE, was increased in the pituitary of LBW rats. Upregulated Gas5 lncRNA may interrupt the glucocorticoid-induced expression of Fkbp5 mRNA. Similarly, downregulation of CRF receptors and POMC expression by activated GR may be impaired in the pituitary of LBW rats alone, leading to prolonged elevation in blood corticosterone levels after stress exposure. Therefore, it was suggested that downregulation of the Gas5 lncRNA level to a normal level may correct the stress responses.

In the present study, stress-exposed corticosterone levels were reversed through nutritional intervention. Although folic acid and vitamin B_12_ levels in the umbilical cord blood of mother rats or in breast milk of lactating rats could not be measured, methyl modulator intervention in the third trimester or in lactating mother rats downregulated the higher levels of Gas5 lncRNA expression in the pituitary and partially restored blood corticosterone levels after stress exposure of LBW rats. However, as shown above, the involvement of DNA methylation in Gas5 lncRNA expression was not clarified.

S-Adenosylmethionine (SAM) metabolized by methyl-modulator intervention becomes a methyl donor of DNA and histone proteins. Saunderson et al. reported that a complex balance among DNA methylation, DNA demethylation, and availability of SAM regulates the induction of genes and their post-stress behavioral responses [38]. They showed that SAM did not affect basal DNA methylation of immediate early genes and their gene expressions. However, administration of SAM prior to stress exposure enhanced DNA methylation at the immediate early gene locus, suppressed gene induction especially in the dentate gyrus, and impaired behavioral immobility after 24 h. These results suggest the difficulty of detecting any changes in DNA methylation in a steady state, and therefore, the alteration may become evident specifically during stimulation. In the future, DNA methylation of Gas5 needs to be investigated in greater detail under various pathological conditions. On the other hand, it has become clear that Gas5 lncRNA interacts with various microRNAs [28,29,30]. From these facts, it is possible that the mechanism in which the expression level of Gas5 lncRNA was increased solely in the pituitary gland of LBW rats may be mediated by microRNA. It is also possible that methyl-modulator interventions may influence their microRNA expression and restore elevated Gas5 lncRNA expression.

As a result, it was found that impaired glucocorticoid responsiveness in LBW rats due to malnutrition during the embryonic period was observed in a pituitary-specific manner. The dampened glucocorticoid response may be related to elevated Gas5 lncRNA expression levels. Prenatal or postnatal methyl-modulator intervention attenuated Gas5 lncRNA expression in the pituitary (though only for a limited period) and reversed elevation of blood corticosterone levels after restraint stress. Elevated blood glucocorticoid levels are associated with a variety of metabolic disorders, including hyperglycemia, hypertension, and susceptibility to infection, as well as psychiatric disorders such as depression. Restoring blood glucocorticoid levels, therefore, can prevent the development of these metabolic or psychiatric disorders. The results of the present study suggest that nutritional interventions in pregnant women with insufficient weight gain, mothers who gave birth to LBW infants, or such infants soon after birth may exert a promising effect on the development of diseases in children.

## 4. Materials and Methods

### 4.1. Animals

Wistar rats were maintained at 23 ± 2 °C with a 12:12-hour light-dark cycle (lights on at 08:00 h, off at 20:00 h). They were allowed ad libitum access to laboratory chow and sterile water. All experimental procedures were reviewed and approved by the Laboratory Animals Ethics Review Committee of Nippon Medical School (#27-067 and #2020-003). All experiments were performed in accordance with relevant guidelines and regulations [39]. We previously generated fetal low-carbohydrate and calorie-restricted rats [40]. Briefly, 18 proestrus female rats (age, 9 weeks) were mated with normal male rats. Dams were housed individually with free access to water and divided into two groups: low-carbohydrate and calorie-restricted diet (LC). Dams were restricted in their calorie intake to 60% of the control group (D08021202, Research Diet Inc., New Brunswick, NJ, USA) during the entire gestational period, whereas control dams freely accessed food during the period. Twelve to twenty pups were obtained from 10 dams of each group. Rat pups born with a body weight of more than 6.0 g, which is the average-2SD body weight of the offspring of normal dams, were excluded. No surrogate mother was used, and 10 rat pups were left at random and raised under the birth mother rat. After weaning, the rats of different litters were mixed.

### 4.2. Dexamethasone Administration

Dexamethasone (1 mg/kg body weight) (ZENOAQ, Nippon Zenyaku Kogyo Co., LTD, Fukushima, Japan) or saline was injected intraperitoneally into male rats (age, 6 weeks). All rats undergo decapitation blood draws between 9 am and 11 am, 24 h after the injections. Their pituitaries, liver, skeletal muscle, and adipose tissues were removed and used for RNA extraction.

### 4.3. Methyl Modulator Supplementation

A methyl modulator diet (#D15090803, Research Diet) was created according to previous reports [41]. As shown in Figure 3, the methyl modulator diet was fed to pregnant mother rats in the third trimester, lactating mother rats immediately after delivery for 3 days, lactating mother rats immediately after delivery for 1 week, and offspring after weaning for 1 week.

### 4.4. Restraint Stress Exposure

Six-week-old rats were wrapped in a flexible wire mesh (12 mm × 12 mm) and kept for 120 min between 09:00 h and 12:00 h in an isolated room. Rats were sacrificed in the adjacent room immediately after restraint, and their trunk blood and anterior pituitaries were collected. Non-stressed control rats were housed in a separate room, sacrificed in the same way, and subjected to identical processing.

### 4.5. RNA Extraction and Real-Time RT-PCR

Then, mRNA and miRNA quantification were performed as previously reported [14]. Total RNA was extracted from abdominal aortas, hearts, kidneys, and pituitaries using RNAiso Plus (Takara, Shiga, Japan). The absorbance of each sample at 260 nm and 280 nm was assayed, and RNA purity was judged as the 260/280 nm ratio (the 260/280 nm ratio of all samples used in this study was higher than 1.7). For miRNA expression analysis, first-strand cDNA was synthesized at 37 °C for 1 h using 500 ng of denatured total RNA and then terminated at 85 °C for 5 min using a Mir-X^®^ miRNA First-Strand Synthesis and SYBR^®^ qRT-PCR kit (Clontech Laboratories Inc., Mountain View, CA, USA). For mRNA expression analyses, first-strand cDNA was generated using 250 ng of denatured total RNA; the reaction mixture was incubated at 37 °C for 15 min, 84 °C for 5 s, and 4 °C for 5 min using a PrimeScript^®^ RT reagent kit with gDNA Eraser (Takara). PCR was performed by denaturation at 94 °C for 5 s and annealing extension at 60 °C for 30 s for 40 cycles using SYBR premix Ex Taq (Takara) and specific primer sets for rat Fkbp5 (RA049269, Takara), GAPDH (RA015380, Takara), or Gas5 lncRNA [14]. To normalize each sample for RNA content, GAPDH, a housekeeping gene, was used for mRNA expression analysis. The 2nd derivative method was used as the standard and for calculating C_t_ values [42].

### 4.6. Measurement of Blood Corticosterone Levels

Aldosterone and corticosterone levels were measured using blood plasma from decapitated rats. Corticosterone was measured using a rat corticosterone ELISA kit (#501320, Cayman Chemical, Ann Arbor, MI, USA).

### 4.7. Statistical Analysis

Unpaired *t*-tests, one-way analysis of variance (ANOVA) followed by Tukey’s multiple comparisons test, or two-way ANOVA followed by Tukey’s multiple comparisons test were used for each statistical analysis. Prism 6.0 software (GraphPad Software, Inc., La Jolla, CA, USA) was used for all calculations. Real-time RT-PCR results are expressed as percentages ± SEM with the control set to 100. A value of *p* less than 0.05 was considered significant.

## Figures and Tables

**Figure 1 ijms-22-09767-f001:**
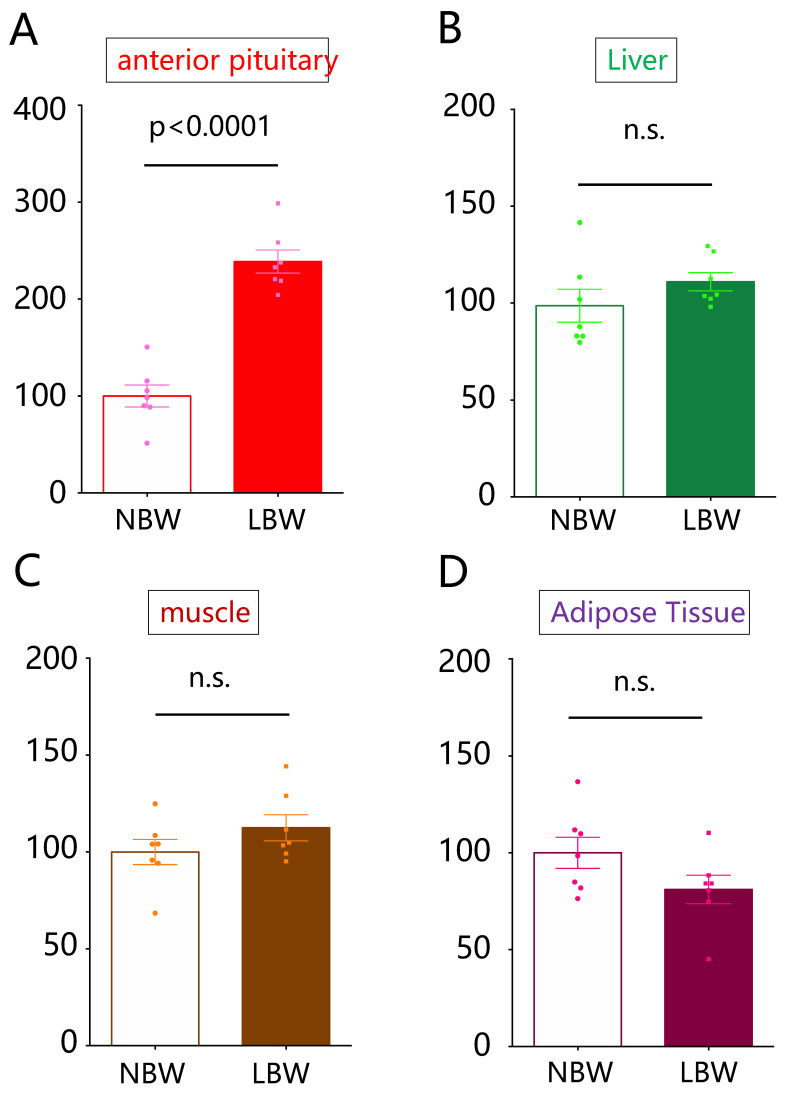
Expressions of Gas5 lncRNA in the pituitary, liver, skeletal muscle, and adipose tissues of NBW and LBW rats. The expression levels of Gas5 lncRNA in the pituitary (**A**), liver (**B**), skeletal muscle (**C**), and adipose tissues (**D**) of control rats (NBW) and LBW rats were quantified. The mRNA expression level is a ratio obtained by correcting the ΔΔCT values of Gas5 lncRNA with the ΔΔCT values of GAPDH and setting the value of NBW to 100 (*n* = 7). Statistical analysis was performed using unpaired *t*-tests. NBW, normal birthweight; LBW, low birthweight, n.s.; not significant.

**Figure 2 ijms-22-09767-f002:**
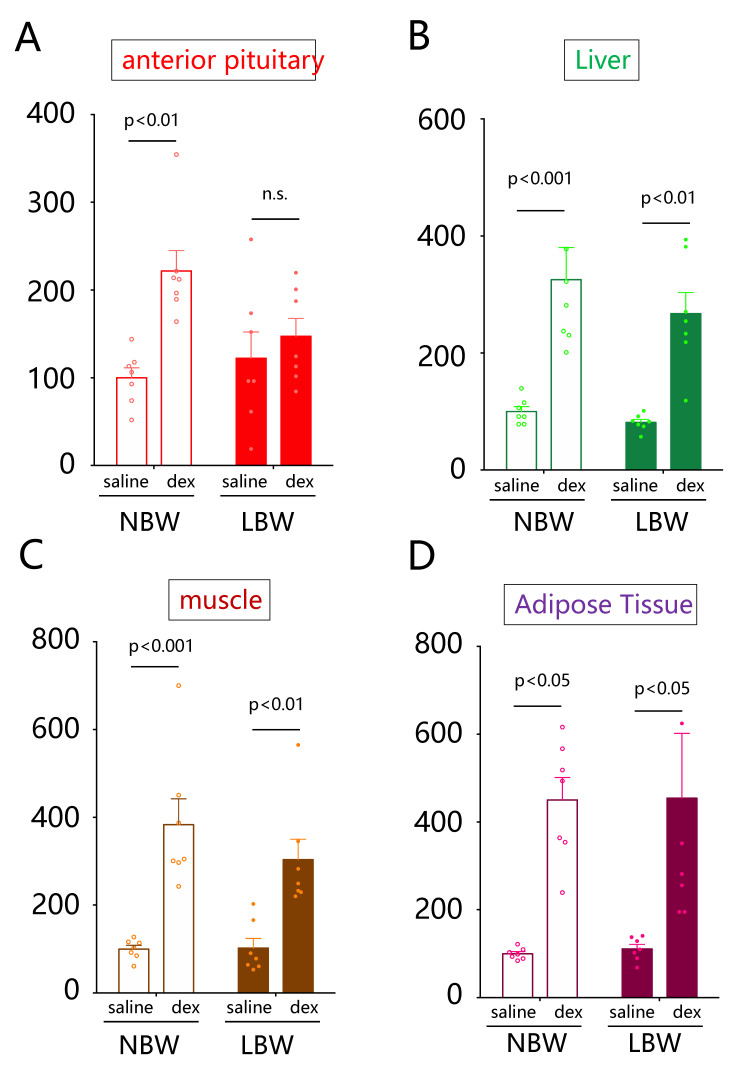
Changes in Fkbp5 mRNA expression after administration of dexamethasone (dex). Dexamethasone (1 mg/kg bodyweight) or saline was injected intraperitoneally into male rats. The rats were killed 24 h after the injections. The mRNA expression levels of Fkbp5 in the pituitary (**A**), liver (**B**), skeletal muscle (**C**), and adipose tissues (**D**) of dexamethasone-administered control rats (NBW) and LBW rats were quantified. The mRNA expression level is a ratio obtained by correcting the ΔΔCT values of Fkbp5 mRNA with the ΔΔCT values of GAPDH and setting the value of NBW to 100 (*n* = 6). Statistical analysis was performed using 2-way ANOVA followed by Tukey’s post hoc test for multiple comparisons. NBW, normal birthweight; and LBW, low birthweight, n.s.; not significant.

**Figure 3 ijms-22-09767-f003:**
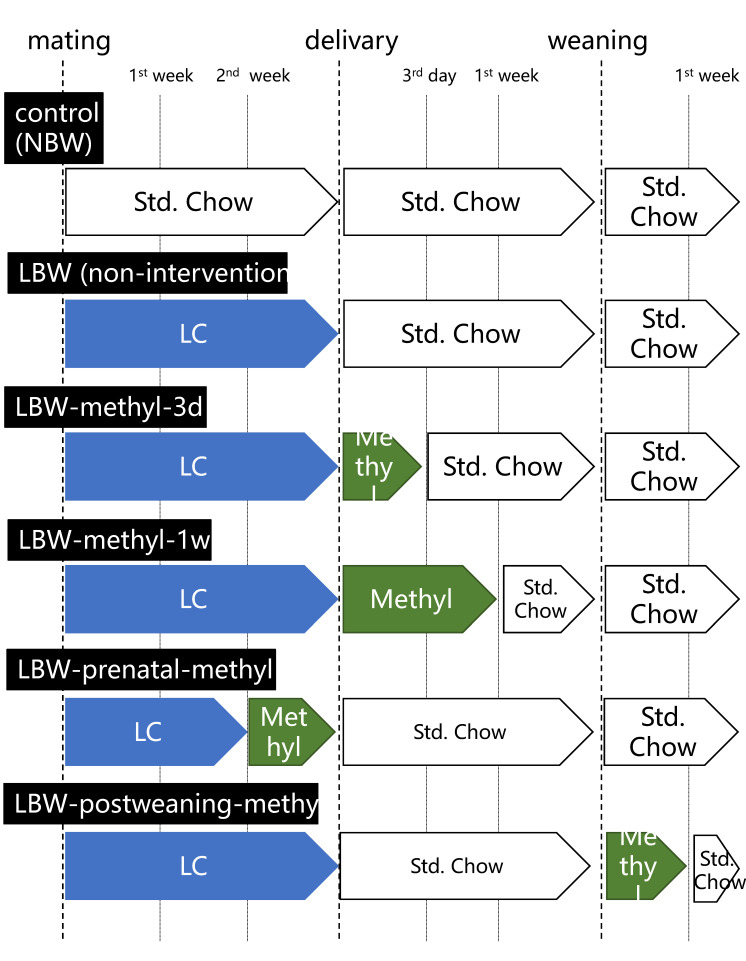
Experimental schema of nutritional intervention. Rats were fed a methyl modulator diet at various times: offspring from the 3-day intervention to the lactating mother rats immediately after delivery (LBW-methyl-3d); offspring from the 1-week intervention to the lactating mother rats immediately after delivery (LBW-methyl-1w); offspring from the 1-week intervention to the mother rats in the third trimester of gestation (LBW-prenatal-methyl); and offspring from the 1-week intervention post-weaning (LBW-postweaning-methyl). NBW, normal birthweight; LBW, low birthweight; Std. Chow, standard chow; LC, low-carbohydrate and calorie-restricted diet; methyl, methyl modulator diet.

**Figure 4 ijms-22-09767-f004:**
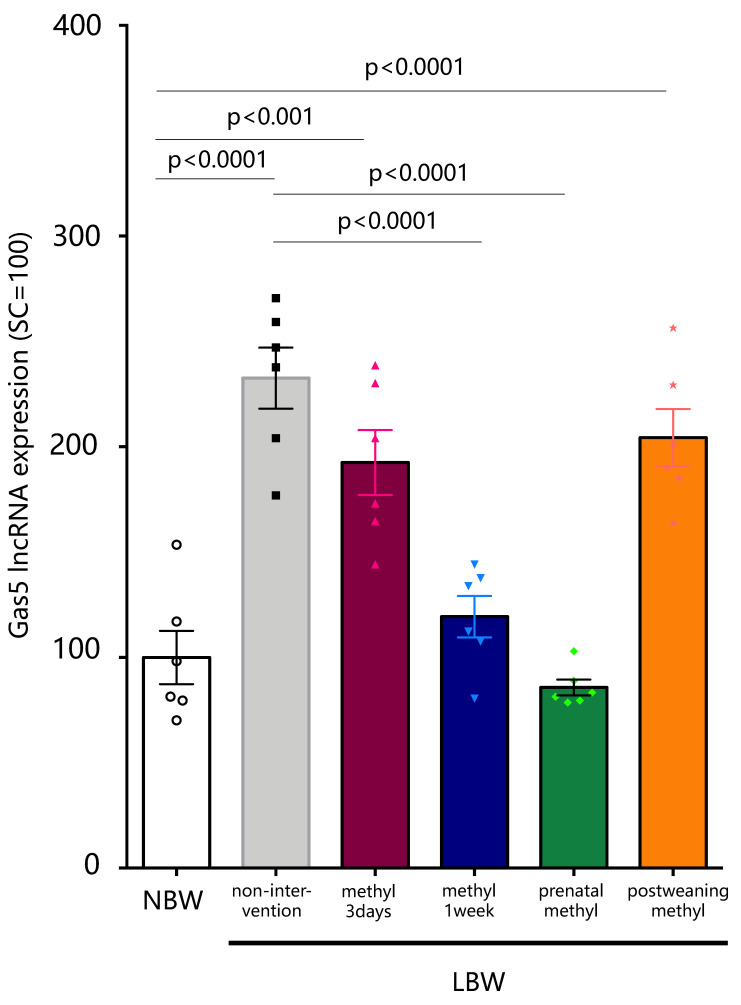
Expression of Gas5 lncRNA in the pituitary of methyl-modulator intervened LBW and NBW rats. The expression levels of Gas5 lncRNA in the pituitary of control rats (NBW) and of LBW rats were quantified. The mRNA expression level is a ratio obtained by correcting the ΔΔCT values of Gas5 lncRNA with the ΔΔCT values of GAPDH and setting the value of NBW to 100 (*n* = 7). Statistical analysis was performed using one-way ANOVA followed by Tukey’s post hoc test for multiple comparisons. Offspring from the 3-day intervention to the lactating mother rats immediately after delivery (LBW-methyl-3d), offspring from the 1-week intervention to the lactating mother rats immediately after delivery (LBW-methyl-1w), offspring from the 1-week intervention to the mother rats in the third trimester of gestation (LBW-prenatal-methyl), and offspring of the 1-week intervention for post-weaning (LBW-postweaning-methyl). NBW, normal birthweight; LBW, low birthweight, n.s.; not significant.

**Figure 5 ijms-22-09767-f005:**
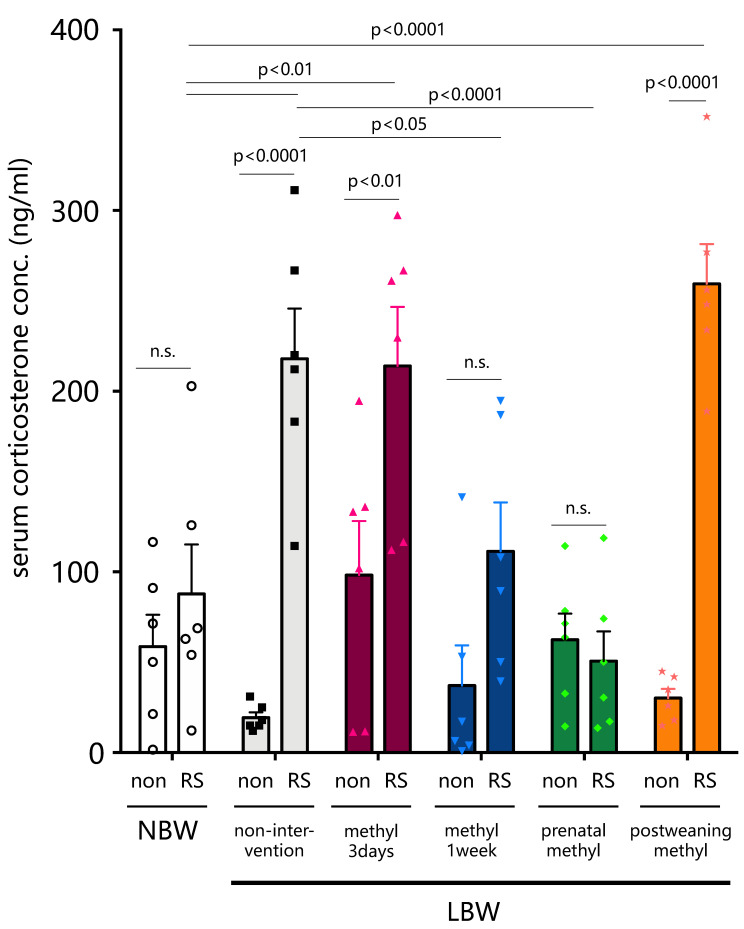
Blood corticosterone levels in restraint stress-exposed (RS) rats with methyl modulator nutritional intervention. Blood corticosterone levels of restraint stress-exposed control rats (NBW) and of LBW rats were measured (*n* = 6). Statistical analysis was performed using 2-way ANOVA followed by Tukey’s post hoc test for multiple comparisons. Offspring from the 3-day intervention to the lactating mother rats immediately after delivery (LBW-methyl-3d), offspring from the 1-week intervention to the lactating mother rats immediately after delivery (LBW-methyl-1w), offspring from the 1-week intervention to the mother rats in the third trimester of gestation (LBW-prenatal-methyl), and offspring from the 1-week intervention post-weaning (LBW-postweaning-methyl). NBW, normal birthweight; LBW, low birthweight, non; non-stressed, n.s.; not significant.

## Data Availability

The raw data supporting the conclusions of this article will be made available by the authors, without undue reservation.

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
