# Peer review of "Prenatal and Postnatal Methyl-Modulator Intervention Corrects the Stress-Induced Glucocorticoid Response in Low-Birthweight Rats"

_ijms, 2021, doi:10.3390/ijms22189767_

Round 1

Reviewer 1 Report

The study would be performed very well.

Author Response

Response: Thank you for your comment on our manuscript. Our manuscript has been confirmed by a native English speaker before submission. This time, when posting the revised manuscript, all parts were checked for spelling and grammar.

Reviewer 2 Report

Nemoto and Kakinuma reported that prenatal and postnatal methyl-modulator intervention corrects the stress-induced glucocorticoid response in low-birthweight (LBW) rats. This is an interesting paper, indicating that the mode of glucocorticoid response in LBW rats is impaired only in the anterior pituitary, and intervention with methyl-modulators ameliorates the impairment with a narrow therapeutic time window.

There are, however, several issues to be addressed to further improve the manuscript.

  1. Which type of the cells that make up anterior pituitary do express Gas5 lncRNA ?
  2. What time did the authors decapitate rats to obtain blood sample for measuring corticosterone?
  3. How did the higher expression level of Gas5 lncRNA affect the brain? Is there any alteration in brain regions involved in depression, anxiety disorder, etc.?

Author Response

  1. Which type of the cells that make up anterior pituitary do express Gas5 lncRNA ?

Response: Unfortunately, we have not established an in situ hybridization experimental system for Gas5 lncRNA, so we have not yet confirmed the cell distribution in the anterior pituitary. However, according to the database, Gas5 lncRNA is ubiquitously expressed in various organs and cell types, so we think it may be expressed in various cells in the anterior pituitary (although there may be differences in its expression level).

https://ngdc.cncb.ac.cn/lncexpdb/gene?geneid=HSALNG0008545

  1. What time did the authors decapitate rats to obtain blood sample for measuring corticosterone?

Response: Our rats are placed under light control every 12 hours (lights up at 8am, turns off at 8pm). All rats undergo decapitation blood draws between 9 am and 11 am. We added the time of decapitation blood sampling to the manuscript (Page 8, Line 293).

  1. How did the higher expression level of Gas5 lncRNA affect the brain? Is there any alteration in brain regions involved in depression, anxiety disorder, etc.?

Response: As I answered in comment # 1, Gas5 lncRNA is expressed in many organs and cells including the brain. Recently, Wu et al. Reported silencing of lncRNA GAS5 could activate PI3K / AKT pathway to protect hippocampal neurons against damage in mice with depression-like behaviors by regulating the miR-26a / EGR1 axis (PMID: 33341564). Their results indicate that Gas5 lncRNA not only acts as a decoy receptor for glucocorticoid receptor but also may inhibit intracellular signal transduction via miRNA. We are planning an experiment to establish an in situ hybridization detection system for Gas5 lncRNA, analyze the expression level of Gas5 lncRNA in the brain in low birth weight rats, and clarify its involvement in neuropsychiatric disorder-like behavioral changes seen in low-birthweight rats. We have added text and citations to the discussion part (Page 7, line 198 – 208 & reference #30).
